# A Systematic Review of Dietary Interventions for Cancer Survivors and Their Families or Caregivers

**DOI:** 10.3390/nu16010056

**Published:** 2023-12-23

**Authors:** Jingle Xu, Rebecca L. Hoover, Nathaniel Woodard, Jennifer Leeman, Rachel Hirschey

**Affiliations:** 1School of Nursing, University of North Carolina at Chapel Hill, Carrington Hall, 120 N. Medical Dr., Chapel Hill, NC 27599, USA; rebec1@email.unc.edu (R.L.H.); jleeman@email.unc.edu (J.L.); hirschey@unc.edu (R.H.); 2Lineberger Comprehensive Cancer Center, University of North Carolina at Chapel Hill, 450 West Dr., Chapel Hill, NC 27599, USA; woodardn@unc.edu

**Keywords:** diet, cancer survivors, behavior change techniques, systematic review, family, caregivers

## Abstract

Family or caregiver engagement has the potential to support healthy dietary changes among cancer survivors. However, little is known about these family- or caregiver-involved dietary interventions and their effects. This systematic review aimed to identify the behavior change techniques (BCTs) used in dietary interventions for cancer survivors and their families or caregivers and to synthesize intervention effects on dietary and health outcomes. Following the PRISMA guidelines, we conducted systematic searches in three databases and identified 12 trials (16 peer-reviewed manuscripts) for inclusion in this review. Data were extracted from these manuscripts and the BCT taxonomy was used to identify the BCTs. A total of 38 BCTs were identified from 12 trials, 13 of which were used in at least half of the 12 trials. Ten studies reported significant intervention effects on health outcomes (e.g., adiposity) and six suggested significant improvements in dietary behaviors (e.g., fruit and vegetable intake). Overall, this review found that family- or caregiver-involved interventions for cancer survivors significantly improved dietary and health outcomes. Future research should identify BCTs particularly for dietary changes and develop effective dyadic strategies to facilitate diet-related interactions between survivors and their families or caregivers to enhance their engagement in healthy diets.

## 1. Introduction

Approximately 32 million individuals live with or beyond a cancer diagnosis (i.e., cancer survivors) worldwide, and the number is steadily increasing [1]. Having a healthy diet can reduce mortality and recurrence risks [2,3,4,5], prevent common comorbidities (e.g., cardiovascular diseases and obesity) [6,7,8,9,10], and improve general health and quality of life [11,12] for cancer survivors. Still, poor adherence to healthy diets is widely observed in this population [13,14,15,16]. Considering the mixed effects of current dietary interventions in this population [17,18], there is a critical need to identify effective strategies to improve the efficacy of dietary interventions for cancer survivors.

Families and caregivers have the potential to encourage and support survivors to adopt and maintain a healthy diet; conversely, family obligations or conflicts in food choices between survivors and their families or caregivers may become a barrier to eating healthy diets [19,20]. Accordingly, interventions have been developed to enhance family or caregiver support or overcome family-related barriers to adopting healthy diets [21,22,23]. An existing scoping review identified the strategies applied in lifestyle interventions for cancer survivors and their families [24]. The scoping review found significant improvements in psychosocial outcomes (e.g., self-efficacy and knowledge) in most of the interventions, with mixed effects on behavioral and health outcomes [24]. Still, no review has systematically identified the behavior change techniques (BCTs) [25] that have been used in family- or caregiver-involved dietary interventions for cancer survivors, as well as the effects of these approaches on dietary and health-related outcomes. BCTs are theory-based “observable, replicable, and irreducible component of an intervention designed to alter or redirect causal processes that regulate behavior” [26] (p. 82). The BCT taxonomy developed by Michie et al. [26], which involves a “hierarchical classification of 93 clearly labeled, well-defined BCTs” (p. 83), is a reliable tool for systematic reviews to extract intervention content across different studies and synthesize and identify effective intervention strategies to guide future research [26]. By identifying the BCTs applied in family- or caregiver-involved dietary interventions for cancer survivors, this review may inform future research on how to improve the efficacy and sustainability of interventions to improve diet quality of cancer survivors and their families or caregivers. Accordingly, this systematic review aims to (1) identify the BCTs reported to be used for modifying dietary behaviors of cancer survivors and their families or caregivers, and (2) synthesize the evidence on the intervention effect on dietary and health-related outcomes.

## 2. Materials and Methods

Following the Preferred Reporting Items for Systematic Reviews and Meta-Analyses (PRISMA) guidelines [27], a systematic review was conducted to identify BCTs and synthesize evidence about the effect of dietary interventions among cancer survivors and their families or caregivers. The review protocol is registered with the PROSPERO International Prospective Register of Systematic Reviews (CRD42023370464).

### 2.1. Search

J.X. conducted searches in three databases, PubMed, CINAHL, and Scopus, from their inception to the last search date of 14 May 2023. J.X. searched keywords and medical subject headings for four concepts: cancer, caregivers, family, and nutrition. The detailed search strings for each database were identified by a librarian with rich experience in health sciences (Appendix A). A total of 1428 articles were found from the three databases after duplicates (*n* = 944) were removed (Figure 1). An additional 44 articles were also identified from reviewing publication reference lists [17,24,28].

### 2.2. Study Selection

The Covidence online platform [29] was used to complete the screening. This review included intervention studies that (1) enrolled adult cancer survivors and their adult family members or caregivers (age ≥ 18 years); (2) had one or more components to improve their diet quality or adherence to healthy dietary behaviors; and (3) reported intervention effects on diet (e.g., diet quality, dietary patterns, dietary behaviors, food intake, or nutrients) or health outcomes. Reports had to be peer-reviewed, original research papers and written in English for inclusion in this review. Reports were excluded if they (1) enrolled childhood cancer survivors; (2) were written in languages other than English; (3) were not peer-reviewed (e.g., thesis, dissertation, a newspaper or magazine article, a website or blog post, or a report published by a government agency or non-governmental organizations); (4) were reviews or systematic reviews; or (5) reported no diet components in the interventions or no dietary or health outcomes. Based on the inclusion and exclusion criteria, three researchers (J.X., R.L.H., and R.H.) reviewed the titles and abstracts of 1428 articles independently. Each report was reviewed by at least two researchers and discrepancies were reconciled through discussion. Two researchers (J.X. and R.L.H.) retrieved the full texts of 119 articles and reviewed them independently. They reconciled discrepancies in full-text screening through discussion and identified 16 reports for inclusion in this review. A flow diagram of the search and screening process is presented in Figure 1.

### 2.3. Data Extraction and Synthesis

J.X. and R.H. developed a structured charting tool to extract data from the 16 included reports. The data charting tool captured characteristics of the study, sample, and intervention along with data on outcomes. Characteristics of the study included study design, purpose, setting, theoretical framework, sample size, and data analysis methods. Information extracted on the sample included sampling strategies, recruitment approach, sample type (survivors and families/caregivers), demographic characteristics (e.g., age, sex, education level, etc.), and cancer-related characteristics (e.g., cancer type, treatment, time since cancer diagnosis, etc.). Intervention information included the intervention content, delivery, frequency, duration, interventionist, and control. Data about dietary and health outcomes and relevant findings were also extracted. One researcher (J.X.) completed the data extraction, and another researcher (R.L.H.) reviewed the 16 included reports and the data matrix table to ensure accuracy of data extraction.

Two reviewers (J.X. and N.W.) applied content analysis [30] to identify BCTs reported in the 16 included manuscripts. J.X. and N.W. completed the BCT taxonomy training [31] and thoroughly read the included manuscripts before coding to be familiar with the codebook and data. J.X. and N.W. used the BCT taxonomy, containing 93 hierarchically clustered BCTs [26], as the codebook to determine the strategies reported to modify dietary behaviors in these studies [26]. Only the BCTs that explicitly aligned with the dietary intervention strategies clearly stated in these reports were identified. Vote-counting and narrative synthesis were used to synthesize the evidence on the BCTs used in these interventions (Aim 1) and intervention effects on dietary and health outcomes (Aim 2).

### 2.4. Quality Appraisal

J.X. and R.L.H. used the study quality assessment tools developed by the National Heart, Lung, and Blood Institute (NHLBI) [32] to assess the quality of the included manuscripts independently and reconciled discrepancies through discussion. The three protocols were excluded in study quality assessment. The items of these tools were scored as yes = 1 and no/not reported (nr)/not applicable (na)/cannot determine (CD) = 0. The quality of the included randomized controlled trials (RCTs) was assessed using the quality assessment tool of controlled intervention studies [32]. The total score of this tool ranged from 0 to 14. RCTs with a score of 11–14 were considered to have good quality, 5–10 fair quality, and 0–4 poor quality [33]. The quality of other intervention studies was assessed using the quality assessment tool for before–after (pre–post) studies with no control group (score range: 0–13) [32]. Other intervention studies with a score of 10–13 were assessed to have good quality, 5–9 fair quality, or 0–4 poor quality.

## 3. Results

After screening, 12 intervention studies (16 reports) were eligible for inclusion in this review. Thirteen were reports of intervention findings and the remaining three reported study protocols. Most articles were excluded either due to no families or caregivers involved in the intervention (*n* = 52) or missing intervention components to improve dietary behaviors (*n* = 27).

### 3.1. Study Characteristics

As presented in Table 1, 6 of the 12 studies were RCTs and 6 were pre-experimental single-group trials. Most studies (*n* = 11) were conducted in the United States. Across the 12 included studies, findings from 703 cancer survivors and 545 family members or caregivers were collectively reported. Most family members or caregivers who participated in these interventions were spouses or intimate partners; three studies only enrolled dyads of cancer survivors and their spouses or intimate partners. Other supportive individuals in these interventions were survivors’ children, friends, siblings, parents, or neighbors. In six studies, researchers recruited participants in dyads. In the remaining six studies, eligible participants were either a cancer survivor or a family member or caregiver to a cancer survivor. Three intervention studies were specifically designed for survivors of underserved racial or ethnic minorities. Across the other nine included studies, most survivors and their families or caregivers were non-Hispanic White. Most of the studies (*n* = 8) included survivors of multiple types of cancer and the remaining four studies targeted a specific type of cancer (breast, *n* = 3; prostate, *n* = 1). More than half of studies only enrolled survivors who had completed their primary cancer treatments (*n* = 7). The average time since cancer diagnosis ranged from less than half a year to more than 10 years.

### 3.2. Quality Assessment

After excluding the three protocols, the quality of 13 reports was assessed. Detailed assessment results are presented in Appendix A. Eight reports had fair quality and five had good quality. All the RCT reports except one had good quality, whose scores ranged from 10 to 12 (*n* = 5). All single-arm pre-experimental studies had fair quality, with a score of 6 or 7. Most risks of bias for RCTs in the current review resided in blinding. The risks of bias for pre-experimental trials in this review resided in sample size, blinding, retention, and multiple outcome measures. Blinding was not recorded or conducted in most of the studies except that outcome accessors were blinded to group assignment in two studies [22,34] and dyads were blinded to group assignment in two studies [21,35]. None of the single-arm pilot trials were powered to achieve a statistical power of 80% or detect clinically meaningful effects. None of the single-arm pilot trials conducted repeated outcome measures before intervention (i.e., interrupted time-series design). One pre-experimental study used semi-structured interviews to collect participants’ experiences of attending an exercise and nutrition program, which reported their pre-to-post changes with no statistical significance determined [23]. 

### 3.3. Intervention Characteristics

Table 2 presents the content and characteristics of the 12 included interventions. Social Cognitive Theory was the theory most frequently used to guide the included studies (*n* = 6). Only three studies integrated dyadic theories, including the Interdependence Theory [47] and the Theory of Communal Coping [48], into the interventions [21,35,42].

No diet-only interventions for cancer survivors and their families or caregivers were found in the current literature. All the included interventions sought to enhance engagement in healthy diets and regular physical activity. Three interventions specifically targeted weight loss/control [21,35,42], one focused on diabetes prevention and management [41], and one aimed to improve ostomy self-management among cancer survivors [44]. 

Across twelve studies, five used a dyad-based approach in which the survivors and families or caregivers were required to be enrolled and received the interventions together and supported each other in changing their behaviors. Seven used an individual-based approach in which they were enrolled in trials and received interventions independently. Two RCTs compared couple-based interventions with individual-based interventions [21,22]. The remaining four RCTs compared the intervention group to an attention control group [34], a waitlist control group [35,37], or a usual care control group [40]. 

In most studies (*n* = 10), specialists (e.g., registered dietitians, exercise physiologists, clinical psychologists, or ostomy nurses) either conducted the interventions directly and/or trained the interventionists. The duration of these interventions ranged from one to twelve months. Ten out of twelve studies conducted baseline and follow-up assessments among both survivors and families/caregivers. In seven of the ten studies, separate effects of these interventions on survivors and their families or caregivers were analyzed, while the remaining three studies combined the data collected among survivors and families/caregivers for analysis. Anton et al. [23] only collected caregivers’ experiences after completing the program. Krouse et al. [44] only assessed the health outcomes of survivors, even though their caregivers were invited to participate in one intervention session.

### 3.4. Behavior Change Techniques

A total of 38 BCTs were identified from these reports (Table 3, Figure 2). Of the 38 BCTs, 13 were identified to be frequently reported in these studies (i.e., reported in at least 6 of the 12 studies). As presented in Figure 2, all 12 studies reported three BCTs, which were the instructions on how to perform the behavior, demonstration of the behavior, and behavioral practice/rehearsal. Most studies also reported problem-solving (*n* = 11), self-monitoring of behavior (*n* = 9), credible source (*n* = 9), unspecified social support (*n* = 7), and information about health consequences (*n* = 7). Half of the studies (*n* = 6) mentioned using goal setting (behavior), goal setting (outcome), feedback on behavior, reduce negative emotions, and restructuring the social environment.

Ten BCTs were reported in some studies (3–5 of the 12 studies; Figure 2). Emotional social support, information about social and environmental consequences, and discrepancies between current behavior and goal were provided by 5 of the 12 studies as BCTs to promote healthy diets. In 4 out of 12 studies, researchers helped survivors and families or caregivers make detailed meal plans (i.e., action planning) [35,40,41,42], set achievable dietary tasks to help survivors and their families or caregivers meet the dietary recommendations (i.e., graded tasks) [21,22,34,35], or provided social reward for their effort or progress in adopting healthy diets [21,35,41,43]. BCTs identified from 3 out of the 12 studies were adding objects to the environment, framing/reframing, self-monitoring of outcome of behavior, and information about antecedents. Demark-Wahnefried et al. (2023) [35], Demark-Wahnefried et al. (2014) [21], and Carmack et al. (2021) [22] directly offered tableware for portion control that could help survivors and families or caregivers adhere to healthy diets (i.e., adding objects to the environment).

The remaining 15 BCTs were rarely reported in these studies (Figure 2). Nine techniques were only used in two of the twelve studies, including review behavior goal(s) [22,43], review outcome goal(s) [22,41], feedback on outcome(s) of behavior [21,42], generalization of target behavior [40,46], and behavior substitution [21,41]. In two of twelve studies, researchers also advised survivors and families or caregivers to evaluate pros and cons of eating healthy diets (i.e., pros and cons) [35,42], minimized the mental resources to help them adopt healthy diets (i.e., conserving mental resources) [35,41], instructed them to avoid exposure to social cues for unhealthy dietary behaviors (i.e., avoidance/reducing exposure to cues for the behavior) [35,42], or prompted them to think about previous successes in eating healthy diets to increase their self-efficacy (i.e., focus on past successes) [35,46]. Six BCTs were only clearly reported in one of the twelve studies, which were information about emotional consequences [23], re-attribution [35], social comparison [21], social support (practical) [43], material reward (behavior) [43], and restructured the physical environment [41]. Table 3 presents typical quotes from the included manuscripts relevant to each BCT.

### 3.5. Dietary Outcomes

#### 3.5.1. Dietary Assessments

A wide range of dietary assessment tools were used to measure dietary outcomes in these studies, including dietary screener questionnaires [34,42,46], food frequency questionnaires [37], 24 h dietary recall [21,22,35], and the diet subscale of a comprehensive health behavior survey [43]. Survivors and families or caregivers were asked to self-report their dietary intake or behaviors via self-administered surveys or when being interviewed by researchers.

#### 3.5.2. Overall Diet Quality/Patterns

In five studies, investigators assessed the overall diet quality or behaviors, three with and two without significant intervention effects. As presented in Table 4, significantly improved diet patterns and nutrition were observed in two single-arm pilot trials [43,46]. Dorfman et al. [42] also found significant improvements in eating behaviors among breast cancer survivors and their intimate partners after the intervention, including cognitive restraint, uncontrolled eating, and emotional eating. No significant between-group differences in diet quality were observed in the two RCTs that assessed overall diet quality [21,35].

#### 3.5.3. Nutrients

Five studies reported nutrients as outcomes, including energy as calculated in calories or Joules (*n* = 3), fat (*n* = 2), and dietary fiber (*n* = 2). Significant intervention effects on calorie [35] and fat intake [22] were revealed in one study for each nutrient and no statistically significant changes in dietary fiber were found in any study (Table 4) [34,37]. No between-group differences were identified in any nutrients. As presented in Table 4, Demark-Wahnefried et al. [35] observed significant decreases in calorie intake in both the intervention and control arms but the decreases between the two groups were not significantly different. Carmack et al. [22] found a significant reduction in daily saturated fat intake among survivors of both the couple-based intervention and survivor-only control groups and a decrease in both total fat and saturated fat consumption among spouses in the intervention group only.

#### 3.5.4. Foods and Drinks

Three studies assessed the intervention effects on the consumption of specific foods or drinks, including fruit and vegetables (*n* = 3), sugar (*n* = 1), red/processed meat (*n* = 1), and alcohol (*n* = 1). As presented in Table 4, two RCTs reported significant within-group changes in fruit and vegetable intake [22,37], one of which found significant between-group differences in changes in vegetable consumption [37]. Survivors and caregivers who received the nutrition and physical activity intervention had significantly greater increases in daily vegetable consumption than those in the waitlist control group [37]. However, the increase in vegetable consumption was not maintained among survivors in the 12-month follow-up reported by Stacey et al. [39]. The other study found no significant differences between the couple-based arm and the survivor-only arm [22]. No statistically significant changes in the consumption of sugar [34], red/processed meat [37], or alcohol [37] were revealed in these studies.

### 3.6. Health-Related Outcomes

#### 3.6.1. Adiposity

Six studies (seven reports) assessed adiposity as an outcome. Adiposity measures used included weight, body mass index (BMI), and waist circumference. All of these studies observed clinically or statistically significant decreases in adiposity between baseline and follow-up assessments (Table 4) [21,22,35,37,39,41,42]. Waist circumference was identified as a more sensitive measure of adiposity than BMI in two studies [21,41]. Between-group differences in weight loss were only observed among survivor–partner dyads by Demark-Wahnefried et al. [35] and the overweight or obese ones by James et al. [37]. Survivors assigned to the individual-based intervention arm had significantly higher reductions in BMI than those in the control group; this significance also applied to waist circumference when comparing the survivor–daughter dyads of team-based or individual-based arms to the control arm, respectively [21]. In the 12-month follow-up of the James et al. [37] program conducted by Stacey et al. [39], only half of the survivors maintained their post-intervention weight loss.

#### 3.6.2. Physical Performance

Five studies reported intervention effects on physical performance. All five studies detected significant effects on at least one measure of physical performance among survivors; this significance did not always apply to their partners or caregivers. Both subjective (e.g., the physical functioning subscale of the medical outcomes short form [49]) and objective measures (e.g., the six-minute walk test (6MWT) [50], the two-minute step test (2MST) [51], and the cardiopulmonary exercise test (VO_2_ peak) [52]) were used across studies. Significantly greater improvements in VO_2_ peak were observed among survivors and their daughters in both the dyad-based intervention arm and individual-based intervention arm in comparison to those of the attention control arm [21]. For survivors, Carmack et al. [22] reported significant within-group changes in different measures of physical functioning between the couple-based and the survivor-only group. Significant within-group improvements in 6MWT and 2MST were observed in the couple-based intervention group, while significant within-group improvements in the 30 s chair stand test (30CST) and the arm curl fitness test (AC) were observed in the survivor-only group. Between-group differences only applied to 30CST and AC in this study [22]. Denmark-Wahnefried et al. [35] reported significantly greater improvements in the sit-and-reach flexibility test (SRT) among survivors and dyads of the intervention group compared to the control group. This study also reported significant within-group changes in 30CST, the eight-foot up and go test (8UG), and SRT among survivors. Caregivers reported better muscular strength, endurance, flexibility, activities of daily living performance, and balance after the exercise and nutrition program in semi-structured interviews [23], which was consistent with the significant within-group increases in the 30CST, SRT, and 2MST among partners observed by Demark-Wahnefried et al. [35]. However, Carmack et al. [22] found no significant within- or between-group changes in physical functioning among spouses of cancer survivors. No significant improvements in physical functioning, as measured by the medical outcomes short form, were reported among either survivors or partners [43]. 

#### 3.6.3. Physical Symptoms

Three studies assessed symptoms as one health-related outcome, with only one study detecting within-group improvements in pain interference among survivors [34]. Crane et al. [34] assessed the summed severity of 15 symptoms using the general symptom distress scale, while Dorfman et al. [42] and Conlon et al. [41] assessed specific symptoms. The medium-to-large effect size for summed symptom severity [34] and significant within-group improvements in pain interference [42] were only detected among survivors; no significant effects were observed among partners or caregivers.

#### 3.6.4. Mental Conditions

Five studies analyzed the changes in mental conditions, including psychological distress [40,42], stress management [43], anxiety and depression [40,44], and general psychological adjustment [40]; all five studies revealed pre-to-post improvements in mental conditions, but significant between-group differences in mental health changes were only found in one study [40]. Caregivers described psychological benefits (e.g., increasing hope and neutralizing helplessness) from the nutrition and exercise programs [23]. Significant within-group improvements in psychological distress [42], stress management [43], and anxiety [44] were observed only among survivors. For couples of shorter relationships (not clearly defined in the original study), Manne et al. [40] observed significant increases in psychological adjustments among spouses who received the healthy lifestyle intervention; the increase was not significant among survivors. Opposite results were observed among couples of longer relationships, where significant increases were found among survivors of the healthy lifestyle intervention group but not among their spouses. Manne et al. [40] also found a significantly higher depression score among survivors with low adherence to traditional masculinity who received the healthy lifestyle intervention than those who received intimacy-enhancing therapy or the usual care, which was not significant among survivors with high adherence to traditional masculinity.

#### 3.6.5. Quality of Life

Six studies, two RCTs, and four single-arm pilot trials measured self-reported, health-related quality of life (QoL) as an outcome; QoL was significantly improved in four studies but not in the remaining two. Surveys used to assess QoL included the short form health survey-36 [21,41,46], the 10-item patient-reported outcome measurement information system (PROMIS) global health survey [35], the functional assessment of cancer therapy (breast) [43], and a four-item health-related QoL framework [44]. Significant increases were found in total QoL among survivors and caregivers [41] and one or multiple sub-domains of QoL among survivors [35,43,44]. 

### 3.7. Psychosocial Constructs

Several studies also measured mediating psychosocial constructs as outcomes, including self-efficacy (*n* = 7), social support (*n* = 3), and barriers to adhering to a healthy diet (*n* = 1); improvements in one or more psychosocial mediators were reported in six studies. Only one study observed significant pre–post increases in diet-related self-efficacy among survivors and caregivers [46]. Improved weight-related self-efficacy was observed only among partners by Dorfman et al. [42] but not among survivors, while symptom self-efficacy significantly increased in both survivors and partners [42]. Knobf et al. [43] and Krouse et al. [44] found significant within-group increases in self-efficacy related to exercise and ostomy management, respectively. Three studies showed no significant impacts on self-efficacy [21,34,35]. Additionally, significantly reduced barriers to adopting a low-calorie diet over time were observed among survivors and dyads [35], with no significant within-group or between-group differences in social support for dietary change [21,35]; however, caregivers who participated in one exercise and nutrition program described that the cohort group format enhanced social support as it provided a platform where caregivers could support each other [23].

## 4. Discussion

This review included 12 studies with dietary interventions targeting adult cancer survivors and their family members or caregivers. Of the 93 BCTs, 38 were used to modify dietary behaviors of cancer survivors and their families or caregivers in the included interventions. The most common BCTs were instructions on how to perform the behavior (*n* = 12), demonstration of the behavior (*n* = 12) or behavioral practice/rehearsal (*n* = 12), problem solving (*n* = 11), self-monitoring of behavior (*n* = 9), and a credible source (*n* = 9). Across the included 12 studies, six interventions induced significant improvements in dietary outcomes (e.g., overall diet quality, eating behaviors, foods, and nutrients) among cancer survivors and families or caregivers. Positive intervention effects on health outcomes (e.g., adiposity, physical performance, and quality of life) were widely observed in ten studies. Some significant improvements in survivors’ dietary and health outcomes were not significant among their families or caregivers. 

This review identified no diet-only interventions in the current literature. As all survivors and their families or caregivers received a combination of dietary and other interventions (e.g., physical activity or symptom management), we cannot conclude that the observed improvements in health outcomes were due to dietary changes alone. The identified 38 BCTs were also often applied to modifying both dietary and physical activity behaviors in these studies. Only six studies showed significant intervention effects on dietary outcomes, including fruit and vegetable consumption [22,37], overall diet quality [43,46], eating behaviors [42], calorie intake [35], and fat intake [22]. This finding is consistent with another systematic review that identified BCTs used in diet and physical activity interventions for colorectal cancer patients, which identified significant effects of these techniques on some dietary behaviors [53]. Thus, there is a critical need for future research to identify effective BCTs and effective methods of applying them to promote healthy diets among cancer survivors and their families or caregivers. For survivors, eating a healthy diet may prevent cancer recurrence [54] and comorbidities [7,55] and improve their quality of life [11,56]. Their families and caregivers also benefit physically and mentally from following a healthy lifestyle, including healthy diets [57,58].

The intervention content identified by this review varied by study but included various education materials, classes or counseling sessions, diet-relevant supplies, and logbooks or website tools to track food consumption. Some interventions identified barriers to maintaining healthy diets and made action plans with survivors and their families or caregivers to stay motivated and maintain the changes beyond the intervention period [35,38,40,41,42,44]. However, a limited number of changes were sustained over an extended period beyond the interventions. A 12-month follow-up assessment of an intervention [37] showed that survivors maintained weight loss but not fruit and vegetable intake [39]. Future dietary interventions should consider other innovative strategies to maintain the beneficial effects beyond the intervention period. For example, a smartphone application may be a convenient tool for real-time tracking of food consumption (i.e., self-monitoring) [59] that survivors, families, and caregivers can continue to use after interventions. Current food-tracking applications recognize food items from a food image and provide individualized food recommendations to users based on their nutritional needs due to the integration of artificial intelligence and computer vision technologies, which greatly increases their convenience and efficacy [59]. With these features, the food-tracking applications have the potential to provide survivors, families, and caregivers with real-time guidance on dietary adjustments beyond the intervention period. 

Although these interventions enrolled both cancer survivors and their families or caregivers, only five studies used dyadic approaches in which dyads of survivors and their families or caregivers were supported together. Two of these five studies compared the effect of dyadic approaches to individual-based approaches; they found no advantageous effects of dyad-based interventions on any dietary or health outcomes in comparison to the individual-based interventions. On the contrary, individual-based approaches led to greater improvements in two physical performance measures than dyadic approaches did among survivors in one study [22]. These findings may indicate that effective approaches are needed to improve the interactions between survivors and families or caregivers to optimize the diet-related support or interventions from external sources, and consequently increase their dietary quality and achieve health benefits. Meanwhile, attempts to modify dietary behaviors may harm their mental health and relationship quality [60,61] by increasing diet-related distress, burden, and interpersonal conflicts [62,63]. This is supported by the finding of Manne et al. [40] that the spouses of prostate cancer survivors who received a nutrition and physical activity intervention had decreased psychological adjustment compared to those receiving the usual care. Future research should gain a deep understanding of how dietary changes and mental health reciprocally influence each other and integrate strategies to alleviate these diet-related adverse mental health effects associated with dietary interventions. Overall, more studies are needed to extend the knowledge about the diet-related interactions among cancer survivors, their partners or caregivers, and external groups (e.g., friends, other family relatives, healthcare professionals, and social media), during the cancer trajectories. Understanding these interactions will facilitate the development of innovative and effective interactive strategies to improve the efficacy of dietary interventions for cancer survivors and their families or caregivers. 

Of the twelve included studies, nine enrolled survivors’ spouses or intimate partners, daughters, or caregivers as intervention participants. The remaining three studies asked cancer survivors to invite a co-survivor [41], partner (not necessarily romantic) [43], or other chosen supportive person [35] to participate in the interventions, whose relationships with survivors were not defined. In three studies, researchers intended to recruit individuals who constantly interacted with survivors and supported them during their cancer trajectories. For example, in one study, eligible co-survivors were required to “act as support for a person with cancer from diagnosis through treatment and beyond” [41]. Denmark-Wahnefried et al. [35] required that the chosen partners interacted with survivors “in person on at least a bi-weekly basis.” No differences in intervention effects were identified by this review between the interventions that involved specific family members or caregivers and those that involved survivor-identified supportive study partners. Consistent with a previous review [64], our review also revealed the knowledge gap of identifying who or what family or caregiving relationships could best support cancer survivors to adopt and maintain healthy diets. Based on the current evidence, we cannot determine whether familism or caregiving relationships, frequent contact and support, or a combination of both better motivates and helps cancer survivors to eat healthy diets. However, numerous observational studies have linked the support from family members or caregivers to survivors’ adherence to healthy diets, especially for those who live with survivors [19,20,65,66,67]. Cancer survivors have also described that family intimacy and responsibilities motivate them to adhere to healthy lifestyle behaviors, including healthy diets, to regain and maintain their health for family members [68]. Hence, a survivor-identified family member or caregiver who frequently contacts and constantly supports cancer survivors may be the best supporter for inclusion in dietary interventions.

Most interventions included in this review implemented lifestyle behavior changes after survivors completed primary cancer treatments; only one study [34] explicitly mentioned that they included survivors receiving active treatment. Post-treatment dietary management is significant for cancer survivors and their families/caregivers because a healthy diet is associated with decreased cancer recurrence [69,70] and comorbidities [6,7,8,9,10] and improved quality of life [11,71]. However, it is also recommended that survivors eat healthy diets as soon as possible after a cancer diagnosis [72]. Thus, future research should focus on developing dietary interventions for those receiving active cancer treatments, such as chemotherapy [73,74]. Dietary interventions during treatment should be tailored to treatment-related challenges and needs. For example, while receiving chemotherapy, patients may experience gastrointestinal issues including nausea [75]. Introducing dietary changes during this time may either be impossible (due to nausea), further impair their appetite, and/or increase mental distress. In this case, easy-to-perform and incremental tasks may be helpful to these survivors, such as focusing on getting nutrients where possible and increasing fruit and vegetable consumption incrementally.

Across the seven studies that tested the intervention effects among survivors and their families/caregivers separately, the interventions had a greater impact on survivors than their families or caregivers. This may be attributed to the family- or caregiving-related barriers. For example, family members and caregivers may neglect their own dietary behaviors [76,77] due to increased burden, time constraints, and elevated emotional distress related to caregiving or shared experiences of cancer. Thus, future interventions should address these barriers to promote healthy diets among survivors’ family members and/or caregivers. While not statistically significant, positive trends were seen in dietary and health outcomes among family members and caregivers. The trend in the results supports the assumption that a cancer diagnosis serves as a “teachable moment” for all to adopt healthy lifestyle behaviors, including healthy diets [78]. 

Only three out of twelve studies were specifically designed for cancer survivors and families or caregivers who came from historically underrepresented backgrounds. There is often low adherence to nutrition recommendations [79,80] and high vulnerability to food insecurity [81] in these communities, which can contribute to adverse health effects [82,83,84]. Families have been shown to effectively shape dietary behaviors among Black and African American [85,86,87] and Hispanic populations [68,88], and as such, integrating family and caregiver support into dietary interventions may be especially effective in promoting health diets in these populations.

This review has the following strengths: (1) The established BCT taxonomy [26] was used to identify intervention strategies reported to modify dietary behaviors among cancer survivors and their families or caregivers. (2) Three researchers reviewed the titles, abstracts, and full texts of the reports to identify those that matched the inclusion criteria. Each report was reviewed by at least two reviewers. (3) The coders (J.X. and N.W.) of the reports using the BCT taxonomy had completed the BCT training. (4) An experienced health sciences librarian determined the search terms used in the three databases. (5) The research team retrieved all the published articles (e.g., protocols and secondary analyses) that reported the 12 included studies from the databases to maximize the information about these interventions from which BCTs could be identified.

However, this review has limitations. Due to the heterogeneity in study populations and outcomes, quantitative synthesis (meta-analysis) could not be used in this review to examine the effects of these interventions on dietary and health outcomes. Furthermore, the research team did not contact the authors of the included reports for additional intervention details and only coded BCTs that were stated in the published descriptions of the interventions. It is possible that included studies used additional BCTs that were not reported clearly enough in these included articles to warrant coding under current BCT definitions.

Overall, in future studies, researchers should continue to optimize this “teachable moment” following cancer treatment to improve dietary behaviors of survivors and their families or caregivers, to help prevent cancer recurrence, reduce comorbidity risks, and improve their quality of life. The interpersonal support and intimate relationships between survivors and families or caregivers may effectively enhance their mutual engagement in healthy diets. Few intervention effects on dietary behaviors and few advantages of dyad-based interventions, in comparison to individual-based interventions, were noted. Thus, future research should gain a deep understanding of the interactions through which cancer survivors and their family members or caregivers influence each other’s dietary behaviors, to develop efficacious approaches, particularly for diet modifications. Finally, to maximize positive and equitable patient outcomes, more dietary interventions during the active treatment period are needed, as well as more interventions focused on the underrepresented populations including, but not limited to, Black/African Americans, Hispanics, and LGBTQ+.

## 5. Conclusions

Overall, various BCTs were used to promote healthy diets in the 12 family- or caregiver-involved dietary interventions reviewed for cancer survivors. Significant intervention effects were widely observed on health outcomes in ten studies, while only six studies suggested significant intervention effects on dietary outcomes.

Future research should focus on developing and testing BCTs and strategies that specifically modify dietary behaviors and are tailored to addressing the interactions among cancer survivors and their families or caregivers. More studies are also needed to develop dietary interventions for survivors receiving active treatments and those of underrepresented populations.

## Figures and Tables

**Figure 1 nutrients-16-00056-f001:**
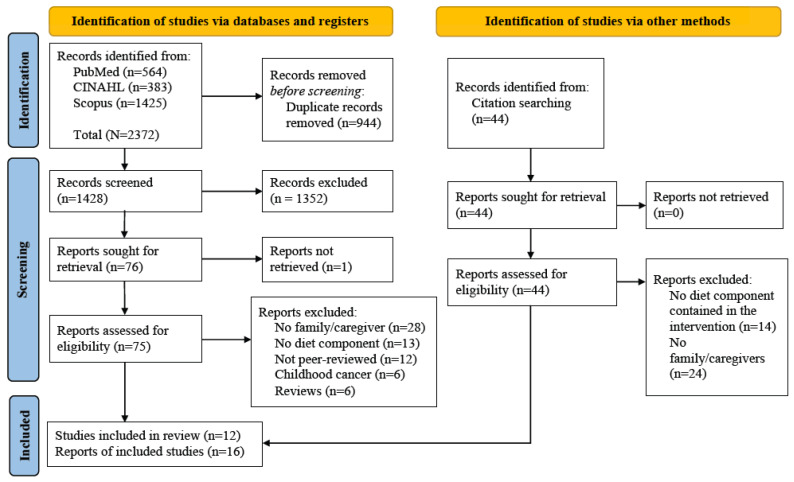
Preferred Reporting Items for Systematic Reviews and Meta-Analyses flow diagram of the search and screening process.

**Figure 2 nutrients-16-00056-f002:**
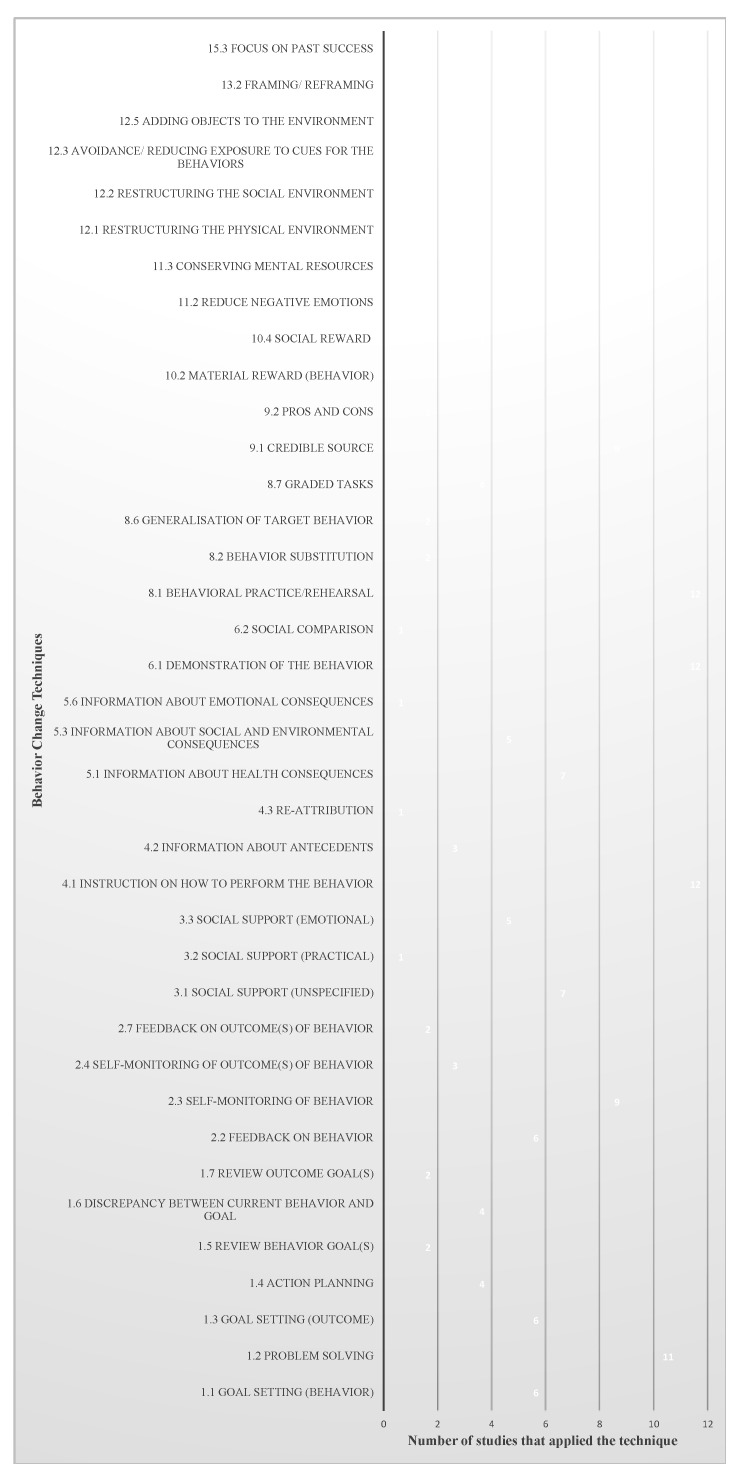
Number of studies that applied each one of the identified 38 behavior change techniques.

**Table 1 nutrients-16-00056-t001:** Study and sample characteristics of the included studies (N = 12).

Author (Year)	Country	*n* (SUR/FC)	Role of FC	Cancer	Time since Diagnosis (Years)	Treatment	Age (SUR/FC)	Gender (M/F)	Race/Ethnicity	Education Level
**Randomized Controlled Trials**										
Carmack (2021) [22]	US	22/22	Spouse	Breast, Prostate, Colorectal	NR	Surgery = 17, Radiation = 11, Hormonal = 11, Chemo = 8, Other = 3	64.1/63.4	21/23	NHW = 32, Hispanic = 7, NHB = 3, Other = 2	High school diploma/GED = 2, Some college = 6, Bachelor’s = 8, Advanced degree = 6,
Crane (2021) [34]	US	45/45	Child = 11, Spouse/partner =10, Friend = 9, Sibling = 4, Parent = 2, Other = 1	Breast, Head or Neck, Liver, Colon, Kidney, Lymphoma, Other	IG 9.02 CG 11.91	Undergoing active cancer treatments = 6	61.16/52.19	0/90	Latina only	<High school = 6, High school/GED = 5, Vocational/technical/community college = 5, 4-year college = 3, Post-graduate = 2
Denmark-Wahnefried (2014) [21]	US	68/68	Daughter	Breast	2	NR	61.3/32.9	0/136	NHW=100, NHB=24, Hispanic White=10, Asian=2	<High school = 1, High school graduate = 25, Some college/junior college/trade school = 49, College graduate = 60
Denmark-Wahnefried (2023) [35] Pekmezi (2021) [36]	US	56/56	Spouse = 23, Friend = 17, Sibling = 7, Child = 6, Other = 3	Breast, Other	5.6	NR	58.4	26/86	NHW = 70, NHB = 41, Other = 1	≤High school = 16, Some college/junior college/trade school = 34, College degree = 59, Unknown = 2
James (2015) [37] James (2011) [38] Stacey (2017) [39]	Australia	96/24 12 both	Spouse/partner = 23, Other relative or friend = 10	Breast, Prostate, Bowel, Colorectal, Melanoma, Other	CG 3.8 IG 3.7	Surgery = 100, Chemo = 73, Radiotherapy = 62, Hormonal = 50	56.6	30/103	NR	Completed post-school qualifications = 95
Manne (2021) [40]	US	237/237	Spouse	Prostate	0.4	Surgery only = 200, Radiation only = 15, Radiation and hormone Rx = 10, Radiation, surgery, and hormone Rx = 2, Surgery and hormone Rx = 8, Surgery and radiation = 2	60.6/57.1	239/235	White = 355, Black = 93, Asian = 5, Hispanic = 11, Other = 7	<High school = 62, Some college = 88, College degree = 114, Above college = 208
**Single-group pre-experimental studies**										
Anton (2013) [23]	US	0/12	Husband = 6, Wife = 3, Girlfriend = 1, Friend = 2	Breast, Uterine, Prostate, Leukemia, Testicular	UN	NR	NA/61.2	6/6	NR	NR
Conlon (2016) [41]	US	66/17	NR	Breast, Gynecological, Lung, Other	5.1	NR	60.5	4/79	NHB = 46, Hispanic/Latino = 22, Other = 15	NR
Dorfman (2022) [42]	US	12/12	Married partner	Breast	1.7	Surgery = 13, Chemo = 8, Radiation = 8, hormonal = 11	58.9/62.7	12/12	White = 20, Black = 4, Non-Hispanic = 23	High school diploma/GED = 5, Some college = 6, Bachelor’s = 11, Master’s = 2
Knobf (2018) [43]	US	35/14	Partner (not necessarily romantic)	Breast	NR	NR	55.7	0/49	Black = 35, White = 1, Hispanic = 4	High school = 8, Technical school = 2, College = 23, Graduate school = 7
Krouse (2016) [44] Grant (2013) [45]	US	38/22	Caregiver	Colorectal, Bladder, Prostate, Ovarian	NR	Colostomy/ileostomy = 23, Urostomy = 11, Unknown = 4	71.3/NA	28/10	NHW = 32, Hispanic White =1, Black = 1	NR
Stoutenberg (2016) [46]	US	16/4	Caregiver	Breast, Multiple, Prostate, Gastric, Myeloma, Pancreatic	NR	NR	62.5	5/15	NR	Some college = 5, Bachelor’s or greater = 15

SUR = survivors, FC = family members or caregivers, US = the United States, M = male, F = female, NHB = non-Hispanic Black, NHW = non-Hispanic White, NA = not applicable, NR= not reported, IG = intervention group, CG = control group.

**Table 2 nutrients-16-00056-t002:** Intervention content and features of the included studies (N = 12).

Author (Year)	DB/ IB	Theory	Interventionist	Frequency	Duration in months	Assessments	Intervention Content	Comparand/Control
**Randomized Controlled Trials**								
Carmack (2021) [22]	DB	Social Cognitive Theory	Counselor	Sessions 1–3 weekly, Sessions 4–5 once every other week, Sessions 6–9 once per month	6	Baseline, 6-month	9 web-based or telephone-based counseling sessions for a couple together; printed tailored workbook and 3 newsletters; guidance, tools, and logbooks to track diet behaviors (couple-based)	No reference to working with a spouse on behavior changes (survivor only)
Crane (2021) [34]	IB	Social Cognitive Theory	Trained bicultural health coach	Once per week	3	Baseline, week 13	20–30 min coaching calls in English or Spanish with dyads separately or together depending on their preferences; Fitbit; symptom management and survivorship handbook	A call from the research team for symptom assessment and change tracking every week
Denmark-Wahnefried (2014) [21]	DB	Social Cognitive Theory, the Transtheoretical Model of Behavior Change, Interdependence Theory, Theory of Communal Coping	NR	Bi-monthly	12	Baseline, 6-month, and 12-month; bi-monthly survey to track progress	Individual group: a personalized initial workbook, 6 tailored newsletters, supplies, and equipment for self-monitoring Dyad group: information and supplies identical to individual group and information to promote effective dyadic communications	Standardized diet and exercise materials
Denmark-Wahnefried (2023) [35] Pekmezi (2021) [36]	DB	Social Cognitive Theory, Interdependence Theory, Theory of Communal Coping	NR	Weekly	6	Baseline, 3-month, and 6-month	A website to provide 24 interactive sessions (15 min each) for feedback and guidance, prompt text messages per week, and daily tips for weight management, diet, and exercise; Portion Doctor tableware; Fitbits; Aria 2 digital scales; and instructions for MyFitnessPal	Waitlist control
James (2015) [37] James (2011) [38] Stacey (2017) [39]	IB	Social Cognitive Theory, Chronic Disease Self-Management Framework	Qualified exercise specialists and accredited practicing dietitian	Weekly and fortnightly	2	Baseline, post-treatment, and 3-month post-treatment (20 weeks)	Walking program, resistance training program, information about healthy eating and maintaining a healthy weight, information delivery and practical activities, workbook, pedometer, Gymstick	Waitlist control
Manne (2021) [40]	DB	Relationship Intimacy Model of Cancer Adaptation	Psychologists, social workers, certified nutritionists, personal trainers	Weekly, booster call 2–3 weeks after	2	Baseline, 5-week, 3-month, and 6-month	5 sessions (90 min) and 1 booster call (30–45 min); Intimacy-Enhancing Therapy intervention arm: couples’ communication regarding cancer, mutual understanding and support, constructive discussion of cancer concerns, emotional intimacy; General Health and Wellness intervention arm: dietary assessment, setting goals, plant-based diet, relaxation	Standard care
**Pre-experimental Trials**								
Anton (2013) [23]	IB	NR	Class instructor, personal trainer	Class twice per week, exercise session twice per week	3	Post-intervention	Basic nutrition and exercise class, individual-tailored exercise session	NA
Conlon (2016) [41]	IB	Bloom’s Taxonomy of Cognitive Development, Social-ecological Framework	Registered dietitian nutritionist, exercise physiologist, or trained staff	Once per week	3 or 1	Pre- to post-program	Culturally and medically adapted nutrition education class (60–75 min) and group exercise class (60 min), a diabetes prevention and management toolkit related to goal setting, food and activity journaling, daily pedometer use, and a “buddy system”	NA
Dorfman (2022) [42]	DB	Interdependence Model of Communal Coping and Behavior Change	Clinical psychologist	Sessions 1–6 weekly, Sessions 7–12 bi-weekly	4.5	Baseline, post-treatment, and 3-month	12 couple-based sessions of weight management, diet and physical activity guidance, appetite awareness training, symptom management protocols, and progress review; written patient manual, Fitbit, food diaries	NA
Knobf (2018) [43]	IB	NR	Exercise physiologist, dietician, oncology nurse practitioner students	Weekly	1.5	Baseline, post-program, 3-month, and 6-month	Face-to-face interactive sessions regarding symptom management, physical activity, healthy eating, bonding, and community sources; prayer at the end of each session	NA
Krouse (2016) [44] Grant (2013) [45]	IB	Chronic Care Model	Research staff, experienced ostomy nurses, peer ostomates	Sessions 1 and 2 on one day, Sessions 3, 4, and 5 one month later	1	Pre-intervention, post-intervention, and 6-month	4 sessions and 1 phone call boost, self-management, social well-being and body image, caregiving, and healthy lifestyle for ostomy	NA
Stoutenberg (2016) [46]	IB	Social Cognitive Theory, the Health Belief Model	Trained facilitator	Weekly	2.5	Baseline, post-program	10 lessons and interactive discussions about general lifestyle activity, resistance training, aerobic activity, general nutrition, cancer nutrition, healthy shopping, weight management, quality sleep, acupuncture and Chinese medicine, and mindfulness	NA

DB = dyad-based, IB = individual-based, NA = not applicable, NR = not clearly reported.

**Table 3 nutrients-16-00056-t003:** Behavior change techniques applied to modify dietary behaviors among cancer survivors and their families or caregivers.

Group	Techniques	Number of Studies	Selected Quotes (Author, Year)
1. Goal and Planning	1.1 Goal setting (behavior)	6	“The behavioral goals were for participants to … consume a diet of ≥7 F and V servings/day for women or ≥9 F and V servings/day for men and ≤7% of total calories from saturated fat” (Carmack et al., 2021) [22] (p. 4)
	1.2 Problem solving	11	“the 3 major foods contributing the highest percentage of kilocalories to each participant’s diet were identified from the dietary recalls performed at baseline … participants were encouraged to … problem-solve on overcoming perceived barriers to healthy behaviors …” (Demark-Wahnefried et al., 2014) [21] (p. 2526)
	1.3 Goal setting (outcome)	6	“[the intervention] promoting a (weight) loss of roughly 0.5 kg per week” (Demark-Wahnefried et al., 2023) [35] (p. 4)
	1.4 Action planning	4	“... and developing an action plan” (Pekmezi et al., 2021) [36] (p. 5)
	1.5 Review behavior goal(s)	2	“...participants were surveyed bimonthly on their progress and plans … The 6 subsequent newsletters provided tailored messages regarding progress toward goals” (Demark-Wahnefried et al., 2014) [21] (p. 2526)
	1.6 Discrepancy between current behavior and goal	4	“...illustrations of current behaviors in relation to national guidelines…” (Carmack et al., 2021) [22] (p. 4) “Count calorie consumption and compare to recommended target.” (Manne et al., 2021) [40] (Appendix A)
	1.7 Review outcome goal(s)	2	“...on Fridays, a “call-to-action” inquired about progress towards incremental goals.” (Demark-Wahnefried et al., 2023) [35] (p. 4)
2. Feedback and Monitoring	2.2 Feedback on behavior	6	“The 6 subsequent letters provide … feedback on portion control …” (Demark-Wahnefried et al., 2014) [21] (p. 2526)
	2.3 Self-monitoring of behavior	9	“... participants were encouraged to keep records of their food intake and physical activity (self-monitoring) …” (Demark-Wahnefried et al., 2014) [21] (p. 2526)
	2.4 Self-monitoring of outcome(s) of behavior	3	“... self-monitoring (through the incorporation of new technologies, i.e., Fitbits and Aria Scales)... Upon randomization, one dyad member was mailed … two sets of instructions to connect to MyFitnessPal^®^ to automate weight …” (Demark-Wahnefried et al., 2023) [35] (pp. 3–4)
	2.7 Feedback on outcome(s) of behavior	2	“… participants were weighed and paper food diaries were reviewed” (Dorfman et al., 2022) [42] (p. 12)
3. Social Support	3.1 Social support (unspecified)	7	“Discuss couple working together to be partners in health and ways to support one another in managing weight and symptoms.” (Dorfman et al., 2022) [42] (p. 11)
	3.2 Social support (practical)	1	“[participants] were encouraged to share any recipes they have tried with the group” (Knobf et al., 2018) [43] (p. 601)
	3.3 Social support (emotional)	5	“Discuss social support and matching support needs with the appropriate support person for managing emotional triggers” (Dorfman et al., 2022) [42] (p. 11)
4. Shaping Knowledge	4.1 Instruction on how to perform the behavior	12	“Each group-based session delivered simultaneous multiple health behavior content covering … information about healthy eating (the Australian Guide to Healthy Eating, fruit and vegetables, maintaining a healthy weight, fats, meat, salt, dietary supplements, alcohol, and food labels).” (James et al., 2015) [37] (p. 4)
	4.2 Information about antecedents	3	“[researchers] have participants identify individual and joint eating triggers.” (Dorfman et al., 2022) [42] (p. 11)
	4.3 Re-attribution	1	“How to recognize hunger, and manage emotional or habitual eating” (Pekmezi et al., 2021) [36] (p. 6)
5. Natural Consequences	5.1 Information about health consequences	7	“Nutrition therapy is recommended for all people with type 1 and type 2 diabetes as an effective component of the overall treatment plan … how eating right and moving more can prevent and control diabetes.” (Conlon et al., 2016) [41] (p. 537)
	5.3 Information about social and environmental consequences	5	“... designed to educate survivors and caregivers on the … importance of basic nutrition and exercise in the management of the many physical and psychosocial issues survivors and caregivers endure” (Anton et al., 2013) [23] (p. 804)
	5.6 Information about emotional consequences	1	“... designed to educate survivors and caregivers on the … importance of basic nutrition and exercise in the management of the many physical and psychosocial issues survivors and caregivers endure” (Anton et al., 2013) [23] (p. 804)
6. Comparison of Behavior	6.1 Demonstration of the behavior	12	“24 weekly interactive sessions averaging 15 min in length were created using Articulate Storyline software (Articulate Global, LLC, New York, NY, USA) to guide participants through topics such as portion control, grocery shopping and food preparation …” (Demark-Wahnefried et al., 2023) [35] (p. 4)
	6.2 Social comparison	1	“Mothers and daughters assigned to the team-based intervention received … information on their other team member” (Demark-Wahnefried et al., 2014) [21] (p. 2526)
8. Repetition and Substitution	8.1 Behavioral practice/rehearsal	12	“... 12-week classes to educate survivors and caregivers on the techniques and importance of basic nutrition and exercise.” (Anton et al., 2013) [23] (p. 804)
	8.2 Behavior substitution	2	“Substituting low-glycemic foods for higher-glycemic load foods…” (Conlon et al., 2016) [41] (p. 537)
	8.6 Generalization of target behavior	2	“After the educational portion of the session, participants were engaged in applied activities, such as … reading nutrition labels to understand calories, sugar, and protein content.” (Stoutenberg et al., 2016) [46] (p. 49)
	8.7 Graded tasks	4	“... each dyad member was encouraged to set incremental goals that would eventually lead over the course of the 6-month intervention …” (Demark-Wahnefried et al., 2023) [35] (p. 4)
9. Comparison of Outcomes	9.1 Credible source	9	“… a presentation on breast cancer among women of color by a Breast Medical Oncologist with a review of common symptoms and symptom management with advanced practice nurses” (Knobf et al., 2018) [43] (p. 601)
	9.2 Pros and cons	2	“Evaluating pros/cons” (Pekmezi et al., 2021) [36] (p. 5)
10. Reward and Threat	10.2 Material reward (behavior)	1	“A variety of jewelry stones, beads, and jewelry making components were available for each participant to create her ‘own significant unique necklace’ as a memorabilia of the (behavioral intervention) experience” (Knobf et al., 2018) [43] (p. 601)
	10.4 Social reward	4	“[participants have] celebration with invited family and guests” (Conlon et al., 2016) [41] (p. 538)
11. Regulation	11.2 Reducing negative emotions	6	“Recognizing how stress influences physical and emotional well-being and strategies to manage it” (Pekmezi et al., 2021) [36] (p. 6)
	11.3 Conserving mental resources	2	“To maintain the pleasure of eating by only limiting food choices when indicated by scientific evidence.” (Conlon et al., 2016) [41] (p. 538)
12. Antecedents	12.1 Restructuring the physical environment	1	“Increase access to affordable, healthy foods in communities, places of work, and schools.” (Conlon et al., 2016) [41] (p. 538)
	12.2 Restructuring the social environment	6	“The survivors and caregivers were encouraged to engage each other in enacting the healthy lifestyle behaviors.” (Crane et al.) [34] (p. 610)
	12.3 Avoidance/reducing exposure to cues for the behaviors	2	“Review the use of assertive communication for managing environmental triggers (specifically making requests and saying no)” (Dorfman et al., 2022) [42] (p. 11)
	12.5 Adding objects to the environment	3	“They also received portion control tableware (Portion Doctor; Portion Health Products, St. Augustine Beach, Fla) … and shoe chips (Nike Inc, Beaverton, Ore) to monitor steps taken, minutes of physical activity, and kilocalories burned.” (Demark-Wahnefried et al., 2014) [21] (p. 2526)
13. Identity	13.2 Framing/Reframing	3	“Learn to refocus and reframe unhelpful thoughts (for weight management goals and wellbeing).” (Dorfman et al., 2022) [42] (p. 11)
15. Self-belief	15.3 Focusing on past success	2	“…such strategies build upon small successes with lifestyle change and thereby enhance self-efficacy” (Pekmezi et al., 2021) [36] (p. 4)

**Table 4 nutrients-16-00056-t004:** Outcomes and findings of the included studies (N = 12).

Author (Year)	Dietary Outcomes			Health Outcomes		
	Survivors	Families/Caregivers	Combined	Survivors	Families/Caregivers	Combined
**Randomized Controlled Trials**						
Carmack (2021) [22]	Positive within-group changes in fruit and vegetable and saturated fat intake in IG and CG	Positive within-group changes in fruit and vegetable, total fat, and saturated fat intake in IG	NA	Positive within-group changes in weight and physical performances in IG and CG	Positive within-group changes in weight in IG	NA
Crane (2021) [34]	Medium-to-large effect sizes for fruit and vegetable total, vegetable only, sugar; medium size for dietary fiber	Medium-to-large effects for total sugar and sugar from sugar-sweetened beverages; medium effect sizes for vegetable intake	NA	Medium-to-large effect sizes for summed symptom severity; small sizes for global symptom distress	NSR	NA
Denmark-Wahnefried (2014) [21]	NSR	NSR	NSR	Greater decreases in BMI and waist circumference in IG than CG	Greater decreases in waist circumference in IG than CG	Greater positive changes in VO_2_ peak and waist circumference in IG than CG
Denmark-Wahnefried (2023) [35] Pekmezi (2021) [36]	Positive within-group changes in calorie intake in IG and CG	Positive within-group changes in calorie intake in IG and CG	Positive within-group changes in calorie intake in IG and CG	Positive changes in weight, waist circumference, physical performance, and physical quality of life in IG and CG; greater improvements in the sit-and-reach test in IG than CG	Positive changes in weight, waist circumference, and physical performance in IG and CG	Positive changes in weight, waist circumference, and physical performance in IG and CG; greater weight loss and improvements in the sit-and-reach test in IG than CG
James (2015) [37] James (2011) [38] Stacey (2017) [39]	NA	NA	Positive within-group changes in the consumption of fruit, vegetables, dietary fiber, fat, and alcohol; between-group differences in changes in daily vegetable consumption between IG and CG; changes not maintained at 12-month follow-up	NA	NA	Positive changes in BMI and weight; between-group differences in weight loss and BMI reduction between IG and CG; changes not maintained in 12-month follow-up
Manne (2021) [40]	NA	NA	NA	Short relationship: positive within-group changes in psychological adjustment in the intimacy-enhancing IG; long relationship: positive within-group changes in intimacy-enhancing and general health and wellness IG	Short relationship: positive within-group changes in psychological adjustment in all groups; long relationship: positive within-group changes in psychological adjustment in the intimacy-enhancing IG and CG	NA
**Pre-experimental Trials**						
Anton (2013) [23]	NA	NSR	NA	NA	Positive within-group improvements in physical performances and psychological outcomes	NA
Conlon (2016) [41]	NA	NA	NSR	NA	NA	Positive within-group changes in waist circumference (12-week) and perceived health (4-week and 12-week)
Dorfman (2022) [42]	Positive within-group changes in eating behaviors	Positive within-group changes in eating behaviors	NA	Positive within-group changes in weight, pain interference, fatigue, symptom self-efficacy, and psychological distress	Positive within-group changes in weight, weight and symptom self-efficacy, and psychological distress	NA
Knobf (2018) [43]	Positive within-group changes in nutrition	Positive within-group changes in nutrition	NA	Positive within-group changes in emotional well-being and stress management	NSR	NA
Krouse (2016) [44] Grant (2013) [45]	NA	NA	NA	Positive within-group changes in health-related quality of life, physical wellbeing, social wellbeing, and anxiety	NA	NA
Stoutenberg (2016) [46]	NA	NA	Positive within-group changes in dietary patterns and self-efficacy of eating habits (sticking to it)	NA	NA	NSR

IG: intervention group; CG: control group/comparand group; NA: not applicable, relevant outcomes were not measured either in the study or in the specific group; NSR: no significant changes reported.

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
