# Peer review of "A Systematic Review of Dietary Interventions for Cancer Survivors and Their Families or Caregivers"

_nutrients, 2023, doi:10.3390/nu16010056_

Round 1
Reviewer 1 Report
Comments and Suggestions for Authors
The review of articles carried out by Xu et al aims to identify the Behavior Change Techniques (BCT) used in dietary interventions for cancer survivors and their families or caregivers and to synthesize intervention effects on dietary and health outcomes. It is a comprehensive review of a topic of interest given the increase in this disease and the need for proper nutritional habits and therefore has great practical application. Here are some tips to improve the quality of this review:
Introduction
- It is possible that some references could be updated, for example those from 2013 or 2014.
- BCT needs to be explained in more depth
Methods
- Indicate from which year to which year the study was screened as well as whether other filters such as provenance, gender or cancer type were added if necessary.
Discussion
- It would greatly improve the understanding of the results to add an explanatory figure to accompany the tables.
Conclusion
- Given the length of the review it is possible that this section may need to be expanded by
- Consider indicating future lines of research or adding another point of practical applications in a separate section line 257 and 264.
It is a very thorough review that provides great insights and if these aspects are improved, it could be published.
Reviewer 2 Report
Comments and Suggestions for Authors
The present study by means of a systematic review has as its main objective to determine the value of diet both in people who have survived cancer and in their families and caregivers.
The topic is of great interest to regular and potential readers of Nutrients.
The systematic review is carried out following all the standards of a work of this type and correctly employing the PRISMA methodology which is the most accepted one.
Three databases (PUBMED, SCOPUS and CINAHL) have been used to obtain the articles on which the study is based, which gives the study greater validity.
The methodology is precisely explained and can be easily replicated by other possible authors.
The results are well defined and can be followed in an adequate and simple manner based on the tables.
The discussion is adequate, comprehensive and supported by an ample bibliography.
The conclusions are in agreement with the proposed objectives.
Recommendations
The obsolescence index of the articles that constitute the bibliography is adequate and is 5 years old.
Some 52% are less than three years old.
Some 22.5% are older than 10 years and could perhaps be updated.
In the tables, in the first column it would be good, if possible, to indicate the quartile to which the article corresponds and whether it is from JCR or SJR.
Reviewer 3 Report
Comments and Suggestions for Authors
This excellent and outstanding systematic review examines clinical nutrition with an emphasis on informal social care, an often overlooked feature in clinical studies. The methodology is sound, the tables are well elaborated, and the discussion is appropriate.
I only have one comment:
This Social Sciences-based work puts a heavy emphasis on the interaction between two people at any time-point. However, often multiple people of multiple (non-medical) backgrounds are involved in the interplay, interacting with each other, without the patient being necessarily present. How did you take this into account?
